# Pharmacokinetics and Tissue Distribution of ^13^C-Labeled Succinic Acid in Mice

**DOI:** 10.3390/nu14224757

**Published:** 2022-11-10

**Authors:** Yonghwan Jung, Jin Sook Song, Sunjoo Ahn

**Affiliations:** 1Therapeutics & Biotechnology Division, Korea Research Institute of Chemical Technology, Daejeon 34114, Republic of Korea; 2Department of Medicinal Chemistry and Pharmacology, Korea University of Science and Technology, Daejeon 34114, Republic of Korea

**Keywords:** succinic acid, LC-MSMS, tissue distribution, pharmacokinetics

## Abstract

Succinic acid is widely used as a food additive, and its effects on sepsis, cancer, ataxia, and obesity were recently reported. Dietary drug exposure studies have been conducted to evaluate the in vivo efficacy of succinic acid, but limited pharmacokinetic information is available. Therefore, this study evaluated the pharmacokinetic profiles and tissue distribution of succinic acid following a single intravenous or oral dose. A surrogate analyte, succinic acid-^13^C_4_ (^13^C_4_SA), was administrated to distinguish endogenous and exogenous succinic acid. The concentration of ^13^C_4_SA was determined by a validated analytical method using mass spectrometry. After a 10 mg/kg intravenous dose, non-compartmental pharmacokinetic analysis in plasma illustrated that the clearance, volume of distribution, and terminal half-life of ^13^C_4_SA were 4574.5 mL/h/kg, 520.8 mL/kg, and 0.56 h, respectively. Oral ^13^C_4_SA was absorbed and distributed quickly (bioavailability, 1.5%) at a dose of 100 mg/kg. In addition, ^13^C_4_SA exposure was the highest in the liver, followed by brown adipose tissue, white adipose tissue, and the kidneys. This is the first report on the pharmacokinetics of succinic acid after a single dose in mice, and these results could provide a foundation for selecting dosing regimens for efficacy studies.

## 1. Introduction

Succinic acid is an intermediate in the citric acid cycle, and its role in energy metabolism is well known. Systemic exposure to succinic acid induces Uncoupling protein 1-dependent thermogenesis, which ameliorates diet-induced obesity and improves glucose tolerance [1,2,3]. In addition, succinic acid has been proposed as a potential treatment for sepsis [4], hematopoiesis [5], and cancer [6,7,8] because of its ability to induce epigenetic changes through its receptor SUCNR1, a G protein-coupled receptor. Short-term exposure to succinic acid is also known to alleviate cerebellar mitochondrial oxidative phosphorylation dysfunction, neurodegeneration, and ataxia [9].

Despite the variety of pharmacological functions of succinic acid, no pharmacokinetic study following intravenous (IV) or oral (PO) administration has been reported. In general, the pharmacological efficacy of succinic acid has been evaluated by in vitro assays or in vivo studies using drinking water or food intake experiments [2,3,9]. In animal studies, it is difficult to evaluate the exact drug consumption for investigating pharmacokinetic/pharmacodynamics (PK/PD) correlations.

Therefore, we assessed the pharmacokinetic profile of succinic acid after a single IV or PO dose and evaluated its accumulation in the heart, liver, kidneys, brain, subcutaneous inguinal white adipose tissue (IWAT), and brown adipose tissue (BAT). To distinguish endogenous and exogenous succinic acid, stable isotope-labeled succinic acid-^13^C_4_ (^13^C_4_SA) was used, and its concentrations were analyzed using ultra high-performance liquid chromatography (UHPLC) with tandem mass spectrometry (MS/MS). Our results lay the groundwork for future in vitro and in vivo studies of the pharmacological effects of succinic acid and provide a basis for selecting optimal dosing regimens for efficacy evaluations.

## 2. Materials and Methods

### 2.1. Chemicals and Reagents

^13^C_4_SA (>98%) and the internal standard (IS) citric acid-2,2,4,4-d4 (CAD_4_, > 98%) were purchased from Cambridge Isotope Laboratories (Andover, MA, USA) and Toronto Research Chemicals (Toronto, ON, Canada), respectively. HPLC-grade methanol (MeOH), acetonitrile (ACN), and water were obtained from Honeywell Burdick & Jackson (Muskegon, MI, USA). Formic acid (99.0%) was purchased from Sam Chun (Pyeongtaek, Korea).

### 2.2. Calibration Standards and Quality Control (QC) Sample Preparation

The initial stock solutions of 1 mg/mL ^13^C_4_SA and CAD_4_ were prepared in MeOH. Calibration standard working solutions and QC working solutions were prepared in MeOH at concentrations ranging from 5 to 80,000 ng/mL. Calibration standards and QC samples were prepared by spiking the aforementioned working solutions with blank plasma or blank tissue homogenate.

### 2.3. Analytical Characterization

The concentrations of ^13^C_4_SA in plasma and tissues were determined using a 1290 Infinity II series UHPLC system (Agilent Technologies, Palo Alto, CA, USA) coupled with an AB Sciex Qtrap 6500+ mass spectrometer (Concord, Canada). A Waters Atlantis Premier BEH C_18_ AX column (2.1 × 100 mm, 1.7 μm particle size) was used, and the column temperature was maintained at 30 °C. The gradient mode consisted of mobile phases A (0.9% formic acid in water) and B (0.9% formic acid in ACN) as follows: 0–1.5 min (0%–0% B), 1.5–4.0 min (5%–30% B), 4.0–4.5 min (30%–30% B), 4.5–5.0 min (30%–0% B), and 5.0–5.5 min (0%–0% B). The flow rate was 0.3 mL/min. Meanwhile, 5-μL prepared samples were injected for analysis. The multiple reaction monitoring mode with negative electrospray ionization was used to quantify ^13^C_4_SA and IS. The source temperature and ion spray voltage were 350 °C and −4500 kV, respectively. The ion source gas 1 flow pressure was 50 psig, and the gas 2 flow setting was 50 psig. The curtain gas pressure was 40 psig, and collision gas is medium. The optimized mass parameters for ^13^C_4_SA and CAD_4_ are listed in Table 1.

### 2.4. Method Validation

The method was validated in terms of specificity, linearity, precision, accuracy, recovery, and matrix effects according to the Guidance for industry: Bioanalytical Method Validation by the Food and Drug Administration [10].

The specificity of the method was evaluated using blank mouse plasma or liver homogenate, ^13^C_4_SA spiked plasma or liver homogenate, and plasma or liver samples after PO administration to exclude potential endogenous interference at the retention times of the analyte and IS.

Linearity was determined by plotting the peak area ratio (y) of samples to IS against the concentrations of the calibration standards (x). The equation was fitted by applying a weight factor of 1/x in linear regression analysis. The lower limit of quantification (LLOQ) was considered the lowest calibration standard, and it could be quantified reliably with acceptable precision of less than 20% and accuracy within ± 20%.

The intra-day precision and accuracy of the method were confirmed by analyzing QC samples at three different concentrations five times on a single day, and the inter-day precision and accuracy were assessed by analyzing the QC samples over 3 consecutive days. For each concentration, five replicates were prepared. Relative standard deviation (RSD) and relative error (RE) were used to express the precision and accuracy, respectively.

The extraction recovery of analytes was assessed by comparing the peak area from five replicate QC samples at low, medium, and high concentrations that were spiked with analytes prior to extraction with the peak area of those that were spiked with blank biological samples. The matrix effects were evaluated by comparing the peak areas of the analytes in post-extracted blank biological samples spiked with QC samples with those of pure standard solutions with the same concentration that were dried directly and reconstituted with the mobile phase. These procedures were repeated for five replicates at three QC concentration levels.

### 2.5. Pharmacokinetic and Tissue Distribution Study

ICR mice (male, 30 ± 5 g, 7–8 weeks old) were obtained from Orient Bio Corporation (Sungnam, South Korea) and housed in an environmentally controlled room for 7 days of acclimatization. The mice were fasted overnight before the day of the experiment. The study was performed under the approval of the Korea Research Institute of Chemical Technology Animal Care and Use committee.

For the pharmacokinetic study, six mice were randomly assigned to two groups (three mice/group) and administered IV 10 mg/kg ^13^C_4_SA or PO 100 mg/kg ^13^C_4_SA. Blood samples were collected by a retro-orbital puncture at the following time points: 0.083, 0.5, 1, 2, 4, 8, and 24 h after IV administration and 0.25, 0.5, 1, 2, 4, 8, and 24 h after PO administration. After centrifugation (15,000 rpm for 3 min at 4 °C), 30-μL plasma samples were mixed with 120 μL of IS solution (500 ng/mL CAD_4_ in 80% MeOH) for protein precipitation. After centrifugation (15,000 rpm for 10 min at 4 °C), 100 μL of the supernatant were transferred to a clean tube and evaporated to dryness in a centrifugal evaporator (EYELA CVE-3100, Tokyo, Japan). The residues were reconstituted with 100 μL of mobile phase A. The calibration samples were prepared by spiking different concentrations of ^13^C_4_SA into the blank plasma.

For the tissue distribution analysis, mice received a single PO dose of 100 mg/kg ^13^C_4_SA. After retro-orbital blood sample collection, the whole body was perfused with saline by placing the perfusion needle into the apex of the left ventricle and cutting the right atrium. Then, heart, liver, kidney, brain, IWAT, and BAT samples were collected and homogenized immediately with IS solution corresponding to three times the tissue weight. After centrifugation (15,000 rpm for 10 min at 4 °C), 120 μL of the tissue supernatant were transferred to a clean tube and evaporated to dryness in a centrifugal evaporator. The residues were reconstituted with 120 μL of the mobile phase. All biological samples filtered using 0.2-μm 96-well AcroPrep Advance filter plates (Pall Corporation, NY, USA) before analysis. Tissue standards covering the expected sample concentration range were made by spiking various concentration of ^13^C_4_SA into the blank tissue homogenate.

The pharmacokinetic parameters of ^13^C_4_SA in plasma were calculated using non-compartmental analysis using Phoenix WinNonlin software version 8.3.4 software. The maximum concentration (C_max_) and time to peak concentration (T_max_) were determined from the concentration–time curve. The terminal rate constant (λ_z_) was calculated by linear regression of the logarithmic plasma concentration versus time curve. The elimination half-life (T_1/2_) was calculated using (ln 2)/λ_z_. The area under the curve from the time of dosing to that of the last measurable concentration (AUC_last_) and the area under the moment curve from the time of dosing to the last measurable concentration (AUMC_last_) were determined using the linear trapezoidal method. The area under the curve from the time of dosing extrapolated to infinity (AUC_∞_) was defined as AUC_last_ + (the last measurable concentration (C_last_)/λ_z_). The area under the first moment curve extrapolated to infinity (AUMC_∞_) was determined as AUMC_last_ + [(time of last measurable observed concentration (T_last_) × C_last_)/λ_z_] + (C_last_/λ_z_^2^). Clearance (CL) was estimated as Dose/AUC_∞_. The mean residence time from the time of dosing to that of the last measurable concentration (MRT_last_) and mean residence time extrapolated to infinity (MRT_∞_) were calculated as AUMC_last_/AUC_last_ and AUMC_∞_/AUC_∞_, respectively. Volume of distribution (V_ss_) was calculated as MRT_∞_ × CL. The absolute oral bioavailability (F) was calculated as (AUC_∞ po_ × Dose_iv_)/(AUC_∞ iv_ × Dose_po_) × 100.

## 3. Results

### 3.1. Method Validation

There was no significant interfering peak observed from endogenous substances in the biological samples at the retention times of ^13^C_4_SA and IS. The typical chromatograms of ^13^C_4_SA and IS were detected using an LC-MS/MS system, as presented in Figure 1.

The regression equation and correlation coefficients (R) for ^13^C_4_SA in different tissue homogenates and plasma are listed in Table 2.

The calibration curve of ^13^C_4_SA in the biological samples exhibited good linearity with R exceeding 0.99. The intra- and inter-day precision and accuracy for ^13^C_4_SA are presented in Table 3.

RSD and RE were typically <15% for all analytes. These data demonstrate that the developed method was reliable and reproducible. The extraction recovery and matrix effect of ^13^C_4_SA are presented in Table 4.

The extraction recoveries of each analyte at different concentrations were consistent, and a little ion suppression was observed in plasma, heart, and liver matrices. The RSD values at middle and high QC levels were <11.7% and the values were <15.5% at LLOQ QC levels in these matrices.

### 3.2. Pharmacokinetics Parameters

The plasma concentration–time profiles after IV injection of 10 mg/kg ^13^C_4_SA and PO administration of 100 mg/kg 13C4SA are presented in Figure 2, and the main pharmacokinetic parameters are listed in Table 5.

After IV administration (10 mg/kg), the plasma concentration of ^13^C_4_SA in mice decreased rapidly in a biexponential manner with T_1/2_ of 0.56 ± 0.09 h. CL of ^13^C_4_SA was 4574.5 ± 744.2 mL/h/kg. V_ss_ and MRT of ^13^C_4_SA were 520.8 ± 88.8 mL/kg and 0.11 ± 0.01 h, respectively. After the PO administration of ^13^C_4_SA, T_max_ was 15 min with C_max_ of 629.7 ± 33.5 ng/mL, which indicated that ^13^C_4_SA was rapidly absorbed. ^13^C_4_SA was rapidly eliminated with T_1/2_ of 0.83 ± 0.21 h. The AUC_last_ for PO and IV ^13^C_4_SA were 321.7 ± 60.6 and 2222.8 ± 349.1, respectively. The calculated F of ^13^C_4_SA was 1.5%. After IV and PO administration, the mean plasma concentration was below the LLOQ at the 4 h time point.

### 3.3. Tissue Distribution

The concentration–time profiles of tissues after a single PO dose of ^13^C_4_SA (100 mg/kg) in mice are summarized in Table 6 and Figure 3.

Following PO administration, ^13^C_4_SA was distributed rapidly in various tissues in mice. T_max_ of ^13^C_4_SA in most tissues was 15 min. The highest C_max_ was observed in the liver (1167.6 ± 183.4 ng/g), followed by BAT (244.8 ± 68.6 ng/g), IWAT (149.0 ± 29.3 ng/g), the kidneys (128.8 ± 47.5 ng/g), the heart (44.5 ± 17.0 ng/g), and the brain (13.3 ± 0.1 ng/g).

## 4. Discussion

Succinic acid, a metabolic intermediate of the citric acid cycle, participates in energy supply, and various pharmacological effects of succinic acid have been reported. The compound improves inflammation and hematopoiesis [5], induces apoptosis in cancer cells [6,7,8], and helps to relieve obesity [2,3]. In general, in vitro or in vivo studies using drinking water or food intake experiments have been conducted to assess the efficacy of succinic acid. However, it is challenging to accurately measure the drug dose in dietary exposure studies. The difficulty in obtaining accurate food intake records has been considered a “fundamental flaw” in human obesity studies [11]. The level of drug exposure is critical for interpreting concentration-dependent drug efficacy, and thus, systemic or target-specific PK/PD analysis can support the optimization of dosing regimens. Therefore, in the present study, the PK profile and tissue distribution of succinic acid were evaluated. To the best of our knowledge, this is the first PK study after a single IV and PO dose of succinic acid in mice.

The first step of this research was developing a rapid and efficient analysis method for the quantitative determination of ^13^C_4_SA. The validated UPLC-MS/MS method required a simple sample preparation procedure and presented good sensitivity (LLOQ < 2 ng/mL) in the biosamples, including mouse plasma and tissues, with a short run time.

To investigate the pharmacokinetic profile of succinic acid, ^13^C_4_SA was administered to ICR mice. The surrogate analyte approach was chosen because of the lack of available succinate-free biosamples in the in vivo study [2,12,13]. The pharmacokinetics of ^13^C_4_SA in mice was characterized by high CL and low-to-moderate V_ss_. After PO administration (100 mg/kg), ^13^C_4_SA was absorbed rapidly and F of ^13^C_4_SA was extremely low (1.5%) in mice under the assumption of linear pharmacokinetics. This might be attributable to poor gastrointestinal permeability and a significant first-pass effect. Further studies to investigate the primary mechanism responsible for the poor F of succinic acid will be conducted in the future.

Following PO administration (100 mg/kg) in mice, the tissues were collected and homogenized. Then, the concentration of ^13^C_4_SA was examined in heart, liver, kidney, brain, IWAT, and BAT samples at seven different time points. ^13^C_4_SA was rapidly distributed in all tissues in mice with T_max_ of 0.25–0.5 h. C_max_ of ^13^C_4_SA was the highest in the liver (1167.6 ng/g), followed by BAT (244.8 ng/g), IWAT (149.0 ng/g), the kidneys (128.8 ng/g), and the heart (44.5 ng/mL). Drug exposure was lowest in the brain (C_max_ < 15 ng/mL). Interestingly, the low exposure was maintained until 24 h after administration, whereas it was eliminated in a multi-exponential manner within 24 h in all other tissues. This suggests that hydrophilic and charged succinic acid does not easily penetrate the blood–brain barrier and it is not metabolized or eliminated across the blood–brain barrier very well. Its accumulation in the brain might be attributable to the limited expression of transporters or exchangers to facilitate its transport. Further research to investigate the factors responsible for the accumulation of succinic acid in the brain might be valuable.

Recently, various therapeutic effects of succinic acid beyond its historical role as a respiratory substrate of the mitochondrial electron transport chain were suggested. However, most in vivo studies that evaluated its efficacy were conducted after drug treatment via dietary exposure, and there was no PK information available. Therefore, the PK profile of succinic acid in mice after a single dose could be beneficial for optimizing dosing regimens for efficacy studies. In addition, drug exposure information in various tissues will help to better understand the role of succinic acid in specific tissues and conduct PK/PD analysis to investigate its effect against specific targets.

## Figures and Tables

**Figure 1 nutrients-14-04757-f001:**
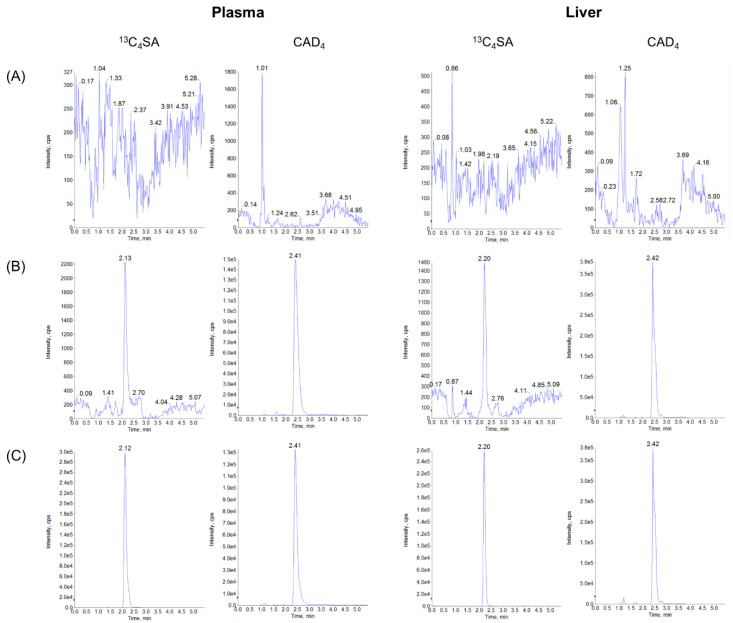
UHPLC-MS/MS chromatograms of ^13^C_4_SA and CAD_4_ in plasma and liver samples: (**A**) blank matrix, (**B**) LLOQ concentration of spiked blank matrices with IS, and (**C**) samples (plasma and liver 15 min after PO administration of 100 mg/kg ^13^C_4_SA).

**Figure 2 nutrients-14-04757-f002:**
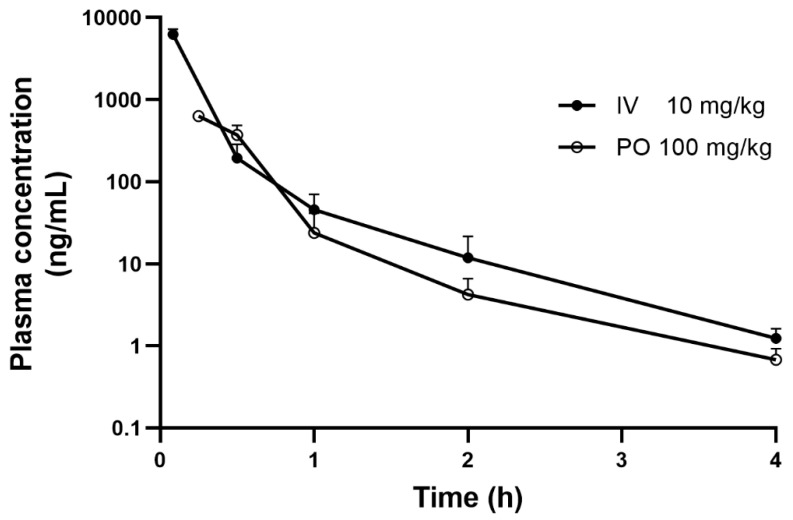
Mean plasma concentration–time curves of ^13^C_4_SA in mice after a single IV and PO dose (mean ± SD, *n* = 3).

**Figure 3 nutrients-14-04757-f003:**
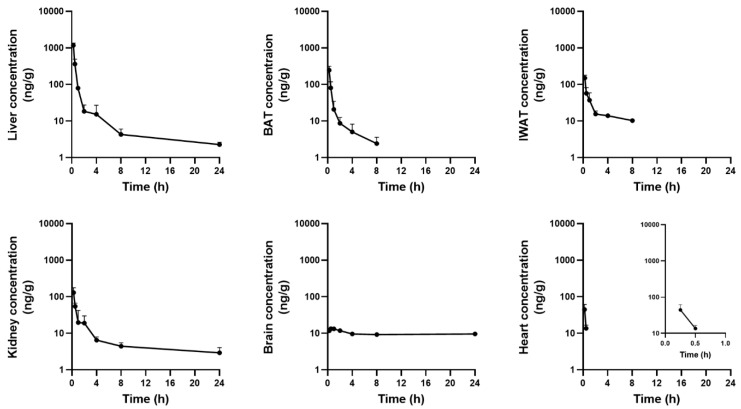
Mean tissue concentration–time curves of ^13^C_4_SA in mice after a PO 100 mg/kg dose (mean ± SD, *n* = 3 per time point).

**Table 1 nutrients-14-04757-t001:** Precursor ion, production ion, Declustering potential (DP), collision energy (CE) and collision exit potential (CXP) of ^13^C_4_SA and CAD_4_.

Analytes	Precursor (m/z)	Production (m/z)	DP (Volts)	CE (Volts)	CXP (Volts)
^13^C_4_SA	120.9	76.1	−20	−16	−9
CAD_4_	194.9	114	−5	−18	−13

**Table 2 nutrients-14-04757-t002:** Equations of ^13^C_4_SA for quantification in plasma and different tissues in mice.

Samples	Standard Curve	R	Linear Range	LLOQ
			(ng/mL)	(ng/mL)
Plasma	0.00096x + 0.000187	0.9999	0.5–8000	0.5
Heart	0.00255x + −0.00384	0.9984	2.0–500	2
Liver	0.00413x + −0.00356	0.9997	0.5–2000	0.5
Kidney	0.00270x + −0.00108	0.9982	0.5–2000	0.5
IWAT	0.00718x + −0.00590	0.9996	2.0–2000	2
BAT	0.00788x + 0.000207	0.9953	0.5–2000	0.5
Brain	0.00634x + −0.01420	0.9946	2.0–2000	0.5

**Table 3 nutrients-14-04757-t003:** Precision and accuracy of ^13^C_4_SA in mouse plasma and tissues (mean ± SD, *n* = 5).

Matrix	Concentration (ng/mL)	Intra-Day			Inter-Day		
		Measured	Precision	Accuracy	Measured	Precision	Accuracy
		(ng/mL)	(RSD, %)	(RE, %)	(ng/mL)	(RSD, %)	(RE, %)
Plasma	0.5	0.50 ± 0.04	8.1	2.8	0.48 ± 0.05	11.1	−3
	500	488.34 ± 6.10	1.3	−2.3	511.24 ± 36.37	7.1	2.3
	8000	7709.59 ± 527.79	6.9	−3.6	7799.37 ± 332.90	4.3	−2.5
Heart	2	1.82 ± 0.08	4.4	−6.6	1.84 ± 0.10	5.2	−5.7
	125	114.21 ± 9.71	8.5	−8.6	121.17 ± 8.71	7.2	−3.1
	500	461.08 ± 36.01	7.8	−7.8	490.41 ± 33.01	6.7	−1.9
Liver	0.5	0.47 ± 0.02	3.6	−4.1	0.49 ± 0.04	9.2	−0.2
	125	126.27 ± 4.80	3.8	1	123.58 ± 6.47	5.2	−1.1
	2000	1971.52 ± 19.63	1	−1.4	1946.28 ± 53.68	2.8	−2.7
Kidney	0.5	0.54 ± 0.07	13	10.3	0.53 ± 0.07	12.8	8.4
	125	121.17 ± 2.71	2.2	−3.1	126.34 ± 7.75	6.1	1.1
	2000	1902.34 ± 61.46	3.2	−4.9	1946.28 ± 53.68	2.8	−2.7
IWAT	2	1.73 ± 0.13	7.5	−11.4	1.83 ± 0.19	10.4	−6.1
	125	141.00 ± 6.66	4.7	12.8	132.63 ± 13.79	10.4	6.1
	2000	2013.07 ± 63.04	3.1	0.7	1978.04 ± 50.92	2.6	−1.1
BAT	0.5	0.48 ± 0.07	14	−1.4	0.53 ± 0.08	14.6	8.2
	125	129.98 ± 6.78	5.2	4	126.34 ± 7.75	6.1	1.1
	2000	2043.73 ± 116.20	5.7	2.2	1946.28 ± 53.68	2.8	−2.7
Brain	0.5	0.50 ± 0.07	14	2.2	0.48 ± 0.06	12.5	−2.1
	125	128.39 ± 7.93	6.2	2.7	128.23 ± 11.63	9.1	2.6
	2000	2910.80 ± 55.73	2.9	−4.5	1963.54 ± 65.03	3.3	−1.8

RSD (%) = (standard deviation of the mean/mean) × 100; RE (%) = [(mean-theoretical concentration)/theoretical concentration] × 100.

**Table 4 nutrients-14-04757-t004:** Recoveries and Matrix effects of ^13^C_4_SA in mouse plasma and tissues (mean ± SD, *n* = 5).

Matrix	Concentration	Extraction Recovery	Matrix Effect
	(ng/mL)	(%)	(%)
Plasma	0.5	89.4 ± 8.0	81.4 ± 8.6
	500	89.1 ± 7.2	97.9 ± 10.3
	8000	86.8 ± 3.9	86.8 ± 3.8
Heart	2	87.0 ± 14.2	88.0 ± 9.4
	125	104.7 ± 5.7	89.5 ± 6.2
	500	97.5 ± 3.1	84.3 ± 8.4
Liver	0.5	105.8 ± 9.2	85.2 ± 13.2
	125	98.5 ± 7.6	83.1 ± 6.2
	2000	90.7 ± 7.0	95.2 ± 9.2
Kidney	0.5	113.5 ± 3.3	89.3 ± 4.0
	125	100.3 ± 9.5	94.9 ± 3.0
	2000	90.1 ± 2.5	93.0 ± 2.3
IWAT	2	86.6 ± 4.3	107.0 ± 2.3
	125	111.8 ± 8.5	95.3 ± 0.7
	2000	104.5 ± 4.5	106.1 ± 2.9
BAT	0.5	97.7 ± 12.8	95.2 ± 3.7
	125	106.7 ± 9.1	97.7 ± 1.5
	2000	91.3 ± 7.6	97.8 ± 6.8
Brain	0.5	95.1 ± 5.1	95.0 ± 11.1
	125	108.0 ± 12.1	99.6 ± 4.0
	2000	103.2 ± 10.4	90.7 ± 1.8

**Table 5 nutrients-14-04757-t005:** ^13^C_4_SA pharmacokinetic parameters in plasma (mean ± SD, *n* = 3).

Pharmacokinetics Parameters	^13^C_4_SA	
	IV	PO
	(10 mg/kg)	(100 mg/kg)
T_max_ (h)	0.08	0.25
C_max_ (ng/mL)	6226.7 ± 994.9	629.7 ± 33.5
T_1/2_ (h)	0.56 ± 0.09	0.83 ± 0.21
AUC_4h_ (ng·h/mL)	2222.8 ± 349.1	321.7 ± 60.6
AUC∞ (ng·h/mL)	2223.8 ± 349.4	322.5 ± 60.3
CL (mL/h/kg)	4574.5 ± 744.2	NA
V_ss_ (mL/kg)	520.8 ± 88.8	NA
MRT_4h_ (h)	0.11 ± 0.01	0.44 ± 0.03
MRT∞ (h)	0.11 ± 0.01	0.45 ± 0.02
F (%)	NA	1.45

NA (Not Applicable).

**Table 6 nutrients-14-04757-t006:** Concentration of ^13^C_4_SA in plasma (ng/mL) and tissues (ng/g) after a single PO administration (100 mg/kg, mean ± SD, *n* = 3).

Time (h)	Plasma	Heart	Liver	Kidney	IWAT	BAT	Brain
0.25	631.0 ± 99.4	44.5 ± 17.0	1167.6 ± 183.4	128.8 ± 47.5	149.0 ± 29.3	244.8 ± 68.6	11.6 ± 0.8
0.5	480.7 ± 45.1	13.6 ± 2.9	360.2 ± 129.8	54.5 ± 11.9	56.9 ± 24.6	80.3 ± 37.4	13.3 ± 0.1
1	46.5 ± 55.6	BQL	78.9 ± 11.4	19.7 ± 22.2	37.1 ± 21.6	20.7 ± 14.0	13.2 ± 1.6
2	3.1 ± 0.2	BQL	18.3 ± 9.0	18.9 ± 10.7	15.5 ± 3.2	8.7 ± 3.8	11.7 ± 1.6
4	0.7 ± 0.2	BQL	15.3 ± 11.6	6.5 ± 1.4	14.9 ± 0.1	5.0 ± 3.2	9.6 ± 0.1
8	BQL	BQL	4.3 ± 1.8	4.4 ± 1.1	12.9 ± 4.79	2.4 ± 1.2	9.2 ± 0.3
24	BQL	BQL	2.3 ± 0.4	2.9 ± 1.1	BQL	BQL	9.6 ± 0.1

BQL (Below the Limit of Quantification).

## Data Availability

Data is contained within the article.

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
