# Peer review of "Pharmacokinetics and Tissue Distribution of 13C-Labeled Succinic Acid in Mice"

_nutrients, 2022, doi:10.3390/nu14224757_

Round 1

Reviewer 1 Report

This manuscript by Jung et al. describes the pharmacokinetic study of succinic acid after a single dose in mice. To evaluate the pharmacokinetic profiles and tissue distribution, the authors administered the 13C4 labeled succinic acid in mice via intravenous and oral doses. The authors have employed UHPLC-MS/MS to analyze the concentrations of isotopically stable succinic acid in mouse plasma and tissues. From their study, the authors found that the oral distribution majorly contributed to the liver, followed by brown adipose tissue, white adipose tissue, and the kidney.

Overall, the current study provided a foundation for selecting dosing regiments for efficacy studies.

Minor comment:

Fig. 2 shows that the plasma concentration-time curves of 13C4 labeled succinic acid in mice after a single IV and PO dose decreased with time. However, the authors mentioned that after PO administration, its plasma concentration increased rapidly. Do the authors have any comments on this?

Author Response

Point 1. Fig. 2 shows that the plasma concentration-time curves of 13C4 labeled succinic acid in mice after a single IV and PO dose decreased with time. However, the authors mentioned that after PO administration, its plasma concentration increased rapidly. Do the authors have any comments on this?

Response1.

We appreciate the comment of the reviewer.

Since the Tmax of succinic acid is calculated as 15 min in this study, we wanted to mention the rapid absorption of succinic acid. Therefore, we changed the ambiguous part as the reviewer suggested (line 188).

Reviewer 2 Report

Dear authors

Here are my comments.

1) Reference 10, should be proper.

2) Matrix effect percentages for certain blank matrix samples were exceeding 20% when considering SD values. Any comments on ion suppression?

3) Does a stable isotope labeled succinic acid absorbs similarly to an endogenous substance?

Author Response

Point 1. Reference 10, should be proper.

Response 1.

We appreciate the comment of the reviewer. We corrected the reference (line 286)

Point 2. Matrix effect percentages for certain blank matrix samples were exceeding 20% when considering SD values. Any comments on ion suppression?

Response 2.

We appreciate the comment of the reviewer. As reviewer mentioned, ion suppression was observed in plasma, heart, and liver matrices. Therefore, we corrected the part and explained the result with the RSD values. (line 176)

Point 3. Does a stable isotope labeled succinic acid absorbs similarly to an endogenous substance?

Response 3.

Since generally a stable-isotope-labeled form of the analyte is known to show the same physicochemical properties than the authentic analyte[1], 13C labeled substances were used in various pharmacokinetic studies [1-4]. Especially, we selected 13C4SA for this study because it was used in several studies to elucidate the mechanism of succinic acid activities [5-7]. Therefore, we added two references to the discussion part (line 229)

  1. Binz, T.M., et al., Development of an LC–MS/MS method for the determination of endogenous cortisol in hair using 13C3-labeled cortisol as surrogate analyte. Journal of Chromatography B, 2016. 1033-1034: p. 65-72.
  2. Toral, P.G., et al., Endogenous synthesis of milk oleic acid in dairy ewes: In vivo measurement using 13C-labeled stearic acid. Journal of Dairy Science, 2017. 100(7): p. 5880-5887.
  3. Moran, N.E., et al., Pharmacokinetics of 13C-Lycopene in Healthy Adults. The FASEB Journal, 2013. 27(S1): p. 38.6-38.6.
  4. Prentice, R.N., et al., A sensitive LC-MS/MS method for the study of exogenously administered 13C-oleoylethanolamide in rat plasma and brain tissue. Journal of Separation Science, 2021. 44(14): p. 2693-2704.
  5. Mills, E.L., et al., Accumulation of succinate controls activation of adipose tissue thermogenesis. Nature, 2018. 560(7716): p. 102-106.
  6. Tannahill, G.M., et al., Succinate is an inflammatory signal that induces IL-1β through HIF-1α. Nature, 2013. 496(7444): p. 238-242.
  7. Hass, D.T., et al., Succinate metabolism in the retinal pigment epithelium uncouples respiration from ATP synthesis. Cell Reports, 2022. 39(10): p. 110917.